# Position: Infringement Cannot Be Cured After Training

**Satoru Utsunomiya** [1]  **Masaru Isonuma** [1 2 3 4]  **Junichiro Mori** [1]  **Ichiro Sakata** [1]

## Abstract

As generative AI faces intensifying legal challenges, the machine learning community has increasingly relied on *post-hoc opt-out*—especially inference-time guardrails and parameter modifications—to argue for compliance. **This paper argues that such post-hoc opt-out methods cannot retroactively cure liability from unlawful data acquisition and training ingestion, because compliance hinges on data lineage, not on outputs.** Our argument has three parts. First, unauthorized copying/ingestion can be a legally complete *completed act*, and model weights may operate as *fixed copies* that retain training-derived expressive value, making later filtering beside the point for infringement. Second, *contract* and *tort/unfair-competition* rules—via licenses, terms of service, and anti-free-riding principles—can independently restrict access and use, often bypassing copyright defenses (e.g., fair use or TDM exceptions). Third, since value from protected inputs can persist in weights, remedies such as *unjust enrichment* and *disgorgement* may require stripping gains and, in some cases, reaching the model itself. We therefore argue for a shift from *Post-Hoc Opt-Out* to verifiable *Ex-Ante Process Compliance*.

## 1. Introduction

The rapid rise of modern foundation models has been enabled by what we might call the data ingestion paradigm: training on unprecedented volumes of text and images drawn from across the open web (Brown et al., 2020; Bommasani et al., 2021). But since these datasets are so large, it has become technically and economically unrealistic to identify and remove every instance of harmful material or protected content before training (i.e., ex-ante filtering). As a result, many developers have proceeded with large-scale training on broadly collected data and have often defended this practice under the banner of fair use.

This approach has triggered serious legal challenges that now threaten the foundations of the AI development ecosystem. Allegations include copyright infringement, privacy violations, and breaches of licensing terms. Rights holders have responded forcefully, arguing that training on protected works and sensitive personal data without authorization is unlawful. High-profile lawsuits—including *The New York Times v. OpenAI* (2023) and *Getty Images v. Stability AI* (2023)[1]—have brought these issues into sharp focus. Importantly, these disputes are not limited to whether model outputs resemble protected works. They also raise broader questions about legal liability for the acts of copying, storing, and using protected or personal data during the training process itself. In this sense, the litigation risk concerns not merely product behavior, but the viability of model development as such.

Against this backdrop—where ex-ante filtering is infeasible and legal pressure is increasing—the research community and industry have increasingly turned toward post-hoc opt-out strategies, namely interventions applied after a model has already been trained. For instance, some generative AIs have introduced opt-out mechanisms intended to prevent the generation of specific copyrighted content, and academic interest in machine unlearning has grown as a technique for removing the influence of particular training data from an already-trained model. These approaches are often presented as pragmatic ways to reduce legal exposure. They are supported by an engineering intuition we call output equivalence: if the system no longer produces infringing or harmful outputs, then it is effectively equivalent to a model that never learned from the problematic data—and therefore should be treated as compliant.

**This paper argues that such post-hoc opt-out strategies do not, as a matter of law, eliminate legal liability arising from the training process.** Across jurisdictions, developers

[1]The University of Tokyo, Tokyo, Japan [2]NII LLMC, Tokyo, Japan [3]Tohoku University, Miyagi, Japan [4]RIKEN, Tokyo, Japan. Correspondence to: Satoru Utsunomiya <utsunomiya-satoru200229@g.ecc.u-tokyo.ac.jp>.

*Proceedings of the 43rd International Conference on Machine Learning*, Seoul, South Korea. PMLR 306, 2026. Copyright 2026 by the author(s).

[1]While liability claims are strong, widespread litigation remains limited by prohibitive legal costs, the emerging nature of individual economic harms, and ongoing uncertainty pending definitive rulings in these foundational cases.

can face multiple, overlapping forms of exposure based on how training data are acquired and used, including *copyright infringement*, *contract-based claims* (e.g., license or terms-of-service violations), and *tort/unfair competition theories*. Because these liabilities can be established based on acts of acquisition, copying, and reproduction—not solely on a model's current output behavior—adjusting outputs after training generally cannot retroactively cure an unlawful training-stage act.

As a result, incurring legal liability can impose substantial costs on developers, potentially resulting in severe remedies such as algorithmic disgorgement and significant monetary exposure. This is why the AI industry must shift away from treating post-hoc opt-out as a compliance strategy and toward verifiable *ex-ante process compliance* (ensuring training data is legally clean and trackable from the start, instead of fixing violations afterward).

In the remainder of this paper, we proceed from context-setting to premises, then to doctrinal analysis, and finally to practical implications. Section 2 surveys related work at the intersection of engineering and law, emphasizing a persistent disconnect between them. Section 3 states and motivates our legal premise by tying unauthorized dataset creation and training to the reproduction right across jurisdictions and showing why key defenses (fair use/TDM) are increasingly constrained. Building on this premise, Section 4 presents our core copyright analysis—introducing the completed-act framing and examining model weights through fixation and perceptibility—showing why suppressing outputs does not retroactively cure training-stage exposure. We then broaden the lens in Section 5 to contract and tort/unfair competition theories that can independently constrain data acquisition and reuse, and in Section 6 to remedial doctrines such as unjust enrichment and disgorgement. Section 7 addresses leading counterarguments and clarifies the limits of defenses commonly invoked in practice. Finally, Section 8 translates our findings into concrete directions for model development and governance.

**Conflict of Interest Disclosure:** The authors declare that they have no competing financial interests or organizational conflicts of interest to disclose.

## 2. Related Work: The Disconnect Between Engineering and Law

Research on legal compliance for generative AI is marked by a clear divide between disciplines. In engineering, most works evaluate technical interventions using engineering metrics while taking legal validity for granted. In legal scholarship, by contrast, much of the discussion focuses on abstract doctrines and policy, often without assessing whether the emerging technical remedies, such as post-hoc

opt-out techniques, are legally meaningful. This section highlights this epistemic gap and our contributions.

### 2.1. Engineering Opt-Out and the Output Equivalence Assumption

The engineering literature has proposed a wide range of techniques intended to suppress or remove undesirable information from trained models. In this paper, we focus on two primary categories of these post-hoc opt-out methods: (1) *inference-time guardrails*, which constrain model behavior without changing the model's internal parameters, and (2) *parameter modification*, which aims to alter the model's internal state so that specific knowledge is removed within the weights themselves (see Appendix B for a detailed technical taxonomy).

Under the latter category of parameter modification, machine unlearning methods—including SISA (Bourtoule et al., 2021) and gradient-based approaches (Jang et al., 2023; Eldan & Russinovich, 2023)—seek to eliminate the influence of specific training examples. Relatedly, model editing techniques such as ROME (Meng et al., 2022) aim to directly modify particular factual associations in model parameters. Conversely, under the former category, inference-time guardrails attempt to prevent infringing outputs through retrieval-based constraints or other runtime controls (Inan et al., 2023).

A central feature of this engineering work is an often implicit assumption we call output equivalence: if post-hoc opt-out make a model's output distribution statistically indistinguishable from that of a counterfactual model trained without the problematic data, then the system is treated as compliant. To ensure the robustness of our legal analysis, we adopt a strong technical premise that represents the best-case performance of post-hoc opt-out methods. Specifically, for the sake of argument, we assume that post-hoc methods such as inference-time guardrails and parameter modification can achieve their theoretical ideal, producing a modified model $f^*$ that is functionally identical to a model retrained from scratch without the infringing data: $f^*(x) = f_{\text{retrained}}(x)$ for any input $x$.

In practice, prior studies emphasize effectiveness and utility metrics (e.g., retention of general capabilities and reduction of targeted behaviors), but they typically do not examine whether even this strongest form of output-level equivalence satisfies the doctrinal requirements of copyright law, particularly with respect to liability arising from upstream acts of ingestion, copying, and reproduction during training.

### 2.2. Legal Scholarship and the Technical Blind Spot

Legal scholarship on LLMs has largely developed along two tracks: (i) debates about the legality of training and (ii)

debates about whether generated content is infringing.

On the input side, scholars generally view training as a non-expressive and transformative use. In the U.S., Lemley & Casey (2021) and Sag (2018) argue that extracting patterns rather than expressive content supports fair use, and Murray (2025) contends more forcefully that AI training broadly qualifies as fair use. In the EU, Christensen (2021) explains how TDM exceptions in the DSM Directive apply, while Dermawan (2024) and Löbling et al. (2024) highlight problems with opt-outs and technical ambiguity. In contrast, Japan's framework is more permissive: Quang (2021) note that non-consumptive training is broadly allowed, offering legal clarity for developers.

On the output side, scholars examine when LLM-generated content might trigger infringement. Lemley (2023) and Sobel (2024) argue that traditional copyright tests—like the idea/expression dichotomy and substantial similarity—struggle to address the nature of generative outputs. In the EU, Rosati (2024) and Dornis & Lucchi (2025) warn that outputs may qualify as unauthorized derivative works, particularly when mimicking style or reproducing prompts.Zhang (2025) and Murray (2023) suggest shifting focus from training legality to regulating output.

Although legal scholarship provides essential doctrine, it rarely asks whether post-hoc opt-out can serve as a legally cognizable cure for training-stage violations. As a result, whether clean-output models remain legally tainted by their data lineage remains largely unresolved. A rare explicit bridge is Marino et al.(Marino et al., 2025), which frames machine unlearning as a potential regulation-aligned compliance tool despite current frictions. In contrast, we argue that for training-stage violations, unlearning and other post-hoc methods cannot, as a matter of principle, retroactively cure the infringement, since liability attaches to unlawful ingestion and copying at training time, not to downstream outputs.

### 2.3. The Significance of Legal-Technical Alignment

The primary contribution of this paper is not merely theoretical but sociotechnical. As long as the engineering and legal communities operate in isolation, the AI industry will remain exposed to various kinds of legal risks. By bridging this epistemic gap, we provide a necessary framework for sustainable AI development, helping to prevent a future where foundational models face existential injunctions due to fundamental misconceptions about liability.

## 3. Legal Baseline: Unauthorized Training as Copyright Infringement

We adopt the following legal baseline: **When copyrighted data are used for model training without authorization,** **the acts of data acquisition and training themselves constitute copyright infringement, particularly by implicating the reproduction right.** We motivate this baseline by examining the definitions and recent judicial scrutiny across jurisdictions. While the contemporary generative AI litigation landscape continues to rapidly evolve under ongoing uncertainty, we explicitly ground our analysis in enduring legal precedents for the foundational principles they establish, ensuring our baseline seamlessly interfaces with modern technological contexts.

### 3.1. Legal Framework: Why Training Is Reproduction and What the Exceptions Require

The threshold question is whether the acts involved in AI training technically constitute reproduction under copyright law. We confirm that they do, based on statutory definitions in major jurisdictions.

- **United States.** 17 U.S.C. (United States Code) §106(1) grants the exclusive right to reproduce a work in copies. A copy is defined in §101 as a fixation in a tangible medium of expression sufficiently permanent to permit it to be perceived or reproduced for more than transitory duration. Exceptions are evaluated under the fair use doctrine (§107), applying four factors: (1) purpose and character of the use, (2) nature of the copyrighted work, (3) amount and substantiality used, and (4) effect on the market.

- **European Union.** Article 2 of the InfoSoc Directive requires member states to protect against direct or indirect, temporary or permanent reproduction by any means and in any form. This broad scope includes digital copying acts. The DSM Directive (Art. 4) introduces a TDM exception, provided that the source is lawfully accessed and rights holders have not opted out through machine-readable means.

- **Japan.** Article 21 of the Copyright Act grants authors the exclusive right to reproduce (*fukusei*), which is defined in Article 2(1)(xv) to include fixation in tangible form, such as server storage. Article 30-4 provides an exception for information analysis, but this is limited by a proviso excluding uses that unreasonably prejudice the interests of the copyright owner.

### 3.2. Vector 1 — Corpus Building: Downloading and Storing as Reproduction

**Establishing the Act: Fixation via Downloading** The act of scraping, downloading, and storing works to build a training corpus constitutes a prima facie violation of the reproduction right. In *Capitol Records v. ReDigi* (2018), the Second Circuit established that creating a new digital file

on a server constitutes a reproduction since a new material object is created. The court reasoned that when a user downloads a file, the user is producing a new phonorecord. Thus, assembling a dataset like The Pile involves creating billions of unauthorized material objects.

**Why Defenses Fail: Illicit Sources and Opt-Outs**  Recent rulings in both the US and EU demonstrate that defenses for this copying are fragile and contingent on strict conditions. In the US, the court in *Bartz v. Anthropic* (2025) denied summary judgment regarding the use of pirated libraries (e.g., Books3), implying that the illicit provenance of data acts as a significant barrier to fair use. If the source material is stolen, the developer's lack of clean hands weighs heavily against excusing the reproduction. Similarly, in the EU, the District Court of Hamburg in *Kneschke v. LAION* (2024) confirmed that downloading images for dataset construction constitutes an act of reproduction under copyright law, even when ultimately covered by a TDM exception. Although the court upheld LAION's reliance on the scientific research exception (§60d UrhG) on the specific facts, it indicated in dicta that for the commercial TDM exception to apply, rights holders' machine-readable opt-outs including reservations expressed in natural-language terms of use must be respected. This reasoning suggests that, outside the narrow research exception, failure to respect such opt-outs would render the copying infringing.

### 3.3. Vector 2 — Training Ingestion: Computational Copies as Reproduction

**Establishing the Act: Fixation via RAM Copies**  The training process requires the repeated loading of datasets into RAM/VRAM. The legal status of these transient yet essential reproductions was addressed in *MAI Systems v. Peak Computer* (1993), where the Ninth Circuit held that loading software into RAM constitutes the creation of a copy. This is because the data is rendered sufficiently permanent to be perceived, reproduced, or otherwise communicated for a period of more than transitory duration. Since AI training involves high-frequency, iterative access to these works over extended periods, these RAM copies satisfy the fixation requirement, establishing a prima facie act of reproduction.

**Why Defenses Fail: The Collapse of Transformative Use**
The legal justification for data training may be weakened under the analysis of the first factor of fair use: the purpose and character of the use. The court's ruling in *Warhol v. Goldsmith* (2023) establishes that a use cannot be considered transformative if it shares the commercial purpose of the original and functions as a market substitute. Generative AI models are fundamentally designed to satisfy the exact same market demand as their training data—providing consumers with expressive text, code, or imagery. Consequently, the

model acts not as a tool for creating new meaning, but as a high-tech substitute that competes for the same user attention. This reality reclassifies the act of ingestion from a creative endeavor into an act of commercial usurpation. Since the ultimate purpose is to generate competing commercial value, we argue the first factor weighs against the developer in the fair use analysis.

## 4. The Incurability of the Infringements

Having established in Section 3 that the acts of dataset creation and training constitute copyright infringements (historical facts), this section argues that post-hoc opt-out techniques cannot retroactively cure these violations. We examine this through the lens of legal doctrines and the ontological status of model weights.

To clarify the scope of our analysis, this section presents two distinct arguments, each targeting a different type of post-hoc opt-out: 4.1 covers all post-hoc methods, regardless of type, whereas 4.2 focuses specifically on inference-time guardrails that restrict model outputs without modifying the model itself.

### 4.1. Completed-Act Doctrine: Liability Attaches at the Moment of Copying

Copyright infringement operates on a principle of strict liability, meaning that the legal wrong is fully consummated the instant an exclusive right is violated. It is not a fluid condition that fluctuates based on the infringer's subsequent remorse or technical patches. Legally, an infringement is a *fait accompli*—an accomplished fact that exists independently of any remedial actions taken later. Once the statutory boundary is crossed, liability attaches permanently to the infringer's record. Consequently, future attempts to hide, delete, or filter the copies are legally irrelevant to the existence of the initial cause of action.

**Dataset Creation as a Completed Fact**  The liability for unauthorized dataset creation is cemented at the exact moment of the initial unlicensed reproduction ($t = 0$). Following the judicial reasoning in *Bartz v. Anthropic* (2025), the mere act of unauthorized possession and reproduction constitutes an independent and complete legal injury. When a developer downloads and saves an unlicensed corpus like Books3 to their server, the statutory violation is finished at that precise second. This legal fact is incurable by subsequent disposition. The law treats this analogously to physical theft: a thief who returns stolen property may reduce their punishment, but they cannot nullify the fact that a theft occurred. Similarly, a developer cannot un-infringe a dataset simply by deleting it or applying a safety filter at a later date ($t = 1$). The snapshot of liability captured at the moment of reproduction remains a permanent part of the legal reality;

post-hoc opt-out are merely attempts to mitigate the scope of damages, not to erase the completed act of infringement.

**Data Training as a Consummated Infringement** The same logic of finality applies with equal force to the training phase. The infringement is perfected the moment the model's weights are updated to reflect the expressive value of the training data. In the context of generative AI, the creation of ephemeral RAM copies serves as the essential mechanism for this value extraction. Once the training process successfully converts the author's protected expression into the developer's commercial neural parameters, the usurpation of the author's labor is done and the point of no return has passed. This process effectively transmutes the right in the expression into a functional component of the model. Once this conversion is realized, the infringement becomes a historical reality that cannot be undone by any kind of update. Subsequent efforts to unlearn specific data or filter outputs are merely post-hoc efforts to suppress the visible symptoms of the infringement; they cannot revert the legal status of the act from completed to uncommitted.

### 4.2. Ongoing Copy Status: Fixation and Recoverability in Model Weights

Under 17 U.S.C. §101, a work qualifies as a copy if it meets two distinct conditions: (1) fixed in a stable medium (Fixation), and (2) retrievable either directly or with the aid of a machine (Perceptibility). We analyze these separately to demonstrate that the post opt-out model remains an infringing copy and that the post-hoc opt-out methods do not negate the underlying embodiment of the work.

**Fixation: Weights Are Stable, Non-Transitory Files** The first inquiry focuses on the stability of the data structure. A work is fixed if its embodiment is sufficiently permanent to exist for more than a transitory duration. In the context of generative AI, model weights constitute static files stored on physical media (e.g., hard drives). Unlike the transient signals of a live broadcast, these weights remain immutable across inference runs unless the model is retrained. Therefore, the weights satisfy the fixation requirement as persistent objects, regardless of whether output filters momentarily suppress their display.

**Machine-Aided Perceptibility: Expression Remains Technically Recoverable** The second inquiry concerns the technical recoverability of the expression. The statute explicitly allows copies to be perceived with the aid of a machine. This means the relevant legal test is not whether a naive user can see the work through a chat interface, but whether the work can be reconstructed by any technical means. Even if safety filters block standard outputs, the expression remains embodied in the weights and re-

coverable through forensic analysis or extraction attacks (Carlini et al., 2023). This aligns with the reasoning in *Micro Star v. FormGen* (1998), where the Ninth Circuit held that user-created MAP files were infringing derivative works because, although mere data files, they functioned as machine-readable instructions that, with the aid of the underlying game engine, recreated protected audiovisual expression. Analogously, model weights serve as machine-readable embodiments from which protected expression can be reconstructed, regardless of whether a secondary filter blocks the final view.

## 5. Beyond Copyright: The Inescapable Web of Contract and Tort Liability

Even if certain forms of training were permissible under the copyright law, developers may still face substantial exposure under *contract* and *tort/unfair competition* doctrines. Explicit licenses and website terms can restrict access and downstream use regardless of copyright exceptions, and unauthorized large-scale appropriation of curated resources may trigger tort or unfair-competition liability. Importantly, the distinction between these liabilities is about the legal basis: because contract claims for violating access terms establish independent grounds of liability that apply even when the data itself is not copyrightable, a developer remains liable for breaching the contractual terms under which they accessed the data even if they establish a copyright defense like fair use. We show that these independent layers of contract and tort liability can constrain training even when copyright defenses might apply, making copyright-based defenses alone insufficient.

### 5.1. Breach of Explicit License Agreements (Signed/Viral Licenses)

A distinct but equally critical vector of liability arises from contract law. When developers ingest data subject to explicit license agreements—such as non-commercial use only or academic license clauses—they bind themselves to specific contractual obligations. Unlike copyright liability, which depends on statutory interpretation, this liability stems from the voluntary breach of a binding promise.

Courts have long upheld that private contracts can impose stricter limitations on data usage than copyright law itself. The seminal case of *ProCD v. Zeidenberg* (1996) provides the definitive precedent. In this case, the Seventh Circuit enforced a consumer-use only license against a defendant who, having purchased a consumer-licensed CD-ROM database, commercially resold its uncopyrightable telephone listings in breach of the license. Crucially, the court held that even where copyright law does not protect the data, the defendant was legally liable for breach of contract. As a result, the court enforced the license terms and remanded the case for

the issuance of a permanent injunction to stop the unauthorized use. In the context of AI, this establishes that a developer who trains a commercial model on research-only data faces strict contractual liability and injunctive relief, regardless of whether the data itself is copyrightable.

### 5.2. Breach of Website Terms of Service (ToS / Browsewrap)

Even without a signed contract, courts have treated enforceable website terms as capable of restricting automated access and bulk copying. This matters because many sites expressly prohibit bots, scraping, or downstream use for model training. Contract law thus enables rights holders to restrict access and use even where copyright defenses might be debatable.

Courts have recognized that private contractual terms can restrict data collection and reuse, effectively overriding the public domain status or statutory exceptions of the underlying material. In *Ryanair v. PR Aviation* (2015), the court held that a database owner could enforce contract terms prohibiting scraping, even if the database itself lacked copyright protection. This principle extends even to jurisdictions often cited as safe havens for AI training, such as Japan. While Copyright Act Article 30-4 broadly authorizes information analysis, courts have clarified that this does not immunize developers from contractual liability for unauthorized access. In *Yomiuri Shimbun v. Digital Alliance* (2005), the Intellectual Property High Court of Japan held that the systematic, commercial, unauthorized use of news headlines created through substantial editorial labor, constituted an actionable tort under Article 709 of the Civil Code, even where copyright protection did not extend to the headlines themselves. This principle applies a fortiori to the violation of a site's explicit prohibition on automated access. Thus, whether in strict or permissive copyright regimes, valid contractual restrictions can block the initial access required to obtain training data, effectively rendering the copyright defense moot.

### 5.3. Tort/Unfair Competition: Liability for Free-Riding on Investment

Where no explicit contract exists, large-scale appropriation of curated resources can still trigger tort-like liability or unfair competition claims. The legal concern is not merely the copying of isolated facts, but the systematic capture of value created by another party's costly investment in collecting, organizing, and maintaining a dataset.

Courts have increasingly recognized that the unauthorized appropriation of such investment constitutes an actionable wrong. In *Tsubasa System v. System Japan* (2001) (Japan) and *CV-Online Latvia v. Melons* (2021) (EU), database scraping was held unlawful as an unfair free-ride on the plaintiff's substantial investment, grounded in tort liability (Civil Code Art. 709) in Japan and in the sui generis database right (Art. 7, Directive 96/9/EC) in the EU. In the AI setting, this doctrine maps cleanly onto the extraction of high-effort corpora—such as professional news archives and specialized legal datasets—whose primary economic value lies in the curation itself. Appropriating this concentrated value to train a commercial model without compensation can therefore be actionable under unfair competition principles, providing a critical layer of protection even when copyright claims remain uncertain.

## 6. No Profiting from Misappropriation: Unjust Enrichment and Disgorgement

The contemporary AI business model often rests on a structural asymmetry: developers can dramatically reduce costs by avoiding licensing or curation expenses, while rights holders receive no compensation. This asymmetry is not only a policy concern but also bears directly on liability and remedies, since unjust enrichment and disgorgement doctrines aim to strip wrongful gains. We therefore examine avoided-cost benefits, head-start advantages, and the limits of post-hoc opt-out as a deterrent.

### 6.1. Illicit Gains from Training Data: Avoided Costs and Head-Start Advantages

The data ingestion paradigm is built on a fundamental economic shortcut: using protected content as computational scaffolding to avoid the immense costs of lawful data acquisition or synthesis.

**High-Value Signals as Avoided Costs** The economic gain from infringement is best measured as avoided cost. Data valuation research shows that AI performance does not scale linearly with volume; rather, it depends disproportionately on a small subset of high-quality signals such as professional journalism or curated books that drive reasoning and reliability (Gunasekar et al., 2023). Legally, the illicit gain is not the retail price of these works, but the massive expenditure saved by bypassing the market to obtain these essential signals. Under *Sheldon v. Metro-Goldwyn Pictures* (1940), disgorgement requires apportioning and surrendering only those profits attributable to the infringement, rather than the infringer's entire revenue. Building on this apportionment principle, we argue that the gain attributable to the wrongful shortcut should include the avoided costs and any incremental commercial advantage causally linked to the infringing data.

**The Head Start Doctrine: Deletion Does Not Remove the Lead** The Head Start doctrine dictates that defendants cannot retain competitive advantages gained through wrongful

shortcuts, even if they attempt later technical fixes. In *Integrated Cash Management Services* (1990), the court recognized injunctive relief as appropriate to neutralize the "head start" attributable to improper use of trade secrets. Likewise, in AI, unlawful corpora materially accelerate model convergence and time-to-market. Because these data points have already performed their functional role in shaping the model's maturity, the developer continues to enjoy a persistent acceleration in the market. Deleting the source files post-training does not neutralize this structural advantage, as the intelligence gained remains in the model's parameters.

### 6.2. Moral Hazard: Why Delete if Caught Is Not an Adequate Deterrent

An enforcement regime that allows companies to keep the principal benefit of wrongful acquisition after paying a manageable penalty risks creating moral hazard. If the expected downside is lower than the cost of licensing, misappropriation becomes a rational strategy. This requires structural relief that extends to the model itself, as seen in precedents where regulators mandated algorithmic disgorgement to prevent firms from retaining the proceeds of unlawful ingestion ((*In re Everalbum*, 2021)).

**The Fallacy of Efficient Breach** If the remedy for wrongful data acquisition is limited to deleting the source files after the model has shipped, the law effectively subsidizes the shortcut. A simple deterrence intuition suggests that a rule requiring only the return of an item if caught invites systematic misconduct. The AI analog is clear: if firms can retain the capability gained from restricted data after deleting the raw dataset, they have effectively laundered the benefit of the violation into a permanent commercial asset. This problem is compounded by the technical limits of trust-based assurances. Once high-value data has been used to train a large model, its influence is distributed throughout a high-dimensional parameter space. Accordingly, a promise to delete the dataset may not meaningfully remove the competitive advantage created by the earlier use, especially where the same training pipeline and derivative checkpoints persist. Without reaching the model itself, the law fails to address the persistent residual advantages of the initial illegality.

## 7. Alternative Views

We acknowledge that the legal landscape regarding AI training remains unsettled, and forceful arguments exist against imposing strict liability. Below, we address three primary counter-arguments often raised by the ML community. While these positions raise valid concerns about innovation and feasibility, we contend that existing legal doctrines ultimately favor a framework of accountability for data ingestion.

That said, concluding that post-hoc methods cannot cure training-stage infringement does not render these techniques meaningless. While they fail to retroactively eliminate liability from the training process, they retain practical value in suppressing infringing outputs and mitigating the scope of monetary damages.

### 7.1. The Dataset Creation as Fair Use Argument

**The View:** Proponents argue that our premise is untenable because training a model is functionally equivalent to human learning—a non-infringing act. Relying on *Authors Guild v. Google* (2015), they contend that creating a training dataset constitutes transformative fair use because the model analyzes statistical patterns rather than engaging in the expressive use of the original works.

**Our Response:** This reliance on the *Google Books* precedent is misplaced as it ignores the fundamental shift in purpose and market harm.

- **From Search to Substitution:** In *Google Books*, the copying was deemed fair because the resulting search index served as a complement to the originals, directing users to them. In contrast, Generative AI datasets are used to create market substitutes. As established in (*Thomson Reuters v. Ross Intelligence*, 2025), copying to build a product that competes with the source material's value is not fair use.

- **The Ingestion Market:** Unlike the search era, a market now exists for licensing data for AI training. By bypassing this market to create datasets for free, developers usurp the copyright holder's right to license their work for computational analysis—a harm recognized in *American Geophysical Union v. Texaco* (1994).

- **The Acquisition Layer Survives:** Independent of the foregoing, even where a court accepts this view, the developer is not rescued. In *Bartz v. Anthropic* (2025), for instance, the court held the training use itself to be transformative, yet it also held that the upstream acquisition of pirated copies remained infringing and was not excused by that transformative training. Our thesis therefore does not depend on how the training step is characterized: liability attaches at acquisition and through the independent contract and tort layers, neither of which a fair use finding on training displaces.

### 7.2. The Data Ingestion as Fair Use Argument

**The View:** Proponents argue that our premise is also untenable since data ingestion is an intermediate step analogous to the reverse engineering in *Sega v. Accolade* (1992), which

was found as a fair use. They contend that intermediate copying is permissible if it is the only way to access unprotected functional elements for legitimate analysis.

**Our Response:** This argument fails because it misapplies the narrow exception in *Sega*, which is limited to intermediate copying necessary to access unprotected functional elements.

- **The Condition of Necessity:** The *Sega* ruling was not a blanket permission. It excused intermediate copying because disassembly was the only way to reach the uncopyrightable functional requirements for interoperability (17 U.S.C. §102(b)), not because the resulting games avoided competition. The exception is contingent on copying unprotected functional elements out of necessity.

- **Market Substitution:** Generative AI does not meet this condition. Rather than copying functional elements out of necessity, AI models ingest protected expression for its expressive value and function as direct market substitutes for their training data. As established in *Thomson Reuters v. Ross Intelligence* (2025), the intermediate copying defense is unavailable when protected expression is copied not out of necessity but to generate products that compete with the original source. This breaks *Sega*'s required nexus, making the ingestion infringing.

### 7.3. The Public Interest and Transaction Costs Argument

**The View:** Finally, it is argued that the transaction costs of licensing billions of documents are prohibitively high. Imposing liability could stifle innovation and bankrupt AI labs, contrary to the public interest. Therefore, copyright exceptions should be expanded to prevent market failure.

**Our Response:** While transaction costs are real, they do not justify a blanket exemption from liability.

- **Licensing Solutions:** High transaction costs are a market design problem, not a legal justification for using data without permission. In practice, collective licensing mechanisms (e.g., the Copyright Clearance Center) exist to reduce this friction. Difficulty in clearing rights does not make unauthorized ingestion legal.

- **Long-term Sustainability:** An AI supply chain built on uncompensated extraction is fragile. As prior work warns about model collapse (Shumailov et al., 2023), training successive generations of models on data that increasingly include model-generated content can lead to a degradation in the quality unless access to original, human-generated data is preserved. Supporting

creators' incentives is therefore aligned with the long-term health of the AI ecosystem.

## 8. Call to Action: From Post-Hoc Opt-Out to Ex-Ante Process Compliance

Our analysis confirms that legal liabilities are established at the exact moment of data acquisition or ingestion. Consequently, post-hoc opt-out techniques (like guardrails and unlearning) are legally insufficient. The industry must undergo a fundamental paradigm shift from cleaning up after the fact to getting it right from the start: a transition to *Ex-ante Process Compliance*. While the goal is to establish a verifiable data lineage before training begins, the responsibilities are shared between those who build the systems and those who set the rules. Importantly, focusing on ex-ante process compliance does not automatically eliminate distinct downstream deployment risks, such as unsafe behaviors learned via in-context learning.

### 8.1. For Engineers (Researchers & Developers): Engineering Verifiability

The academic and development communities must abandon the black box approach. Instead of relying on trust or probabilistic unlearning, they must design architectures that allow for mechanical verification of data hygiene. Critically, these components must operate as a unified system: the "Glass Pipeline" acts as a diagnostic tool to precisely detect when unauthorized examples are ingested, while "Architectural Reversibility" provides the mechanism to surgically undo that specific training step. We articulate these complementary frameworks below:

- **Cryptographic Transparency (The Glass Pipeline):** To balance secrecy with accountability, developers can build a "Glass Pipeline" using cryptographic tooling. Using standards such as C2PA (Coalition for Content Provenance and Authenticity, 2023), each weight update can be tied to verifiable provenance credentials from rights holders (positive verification). Complementarily, Zero-Knowledge Proofs (ZKPs) can enable auditors to verify that restricted works were excluded from training without revealing the full corpus (negative verification). Together, these mechanisms replace trust-based assurances with auditable compliance claims.

- **Architectural Reversibility (Git for Models):** Recognizing that knowledge from training data cannot always be cleanly removed once it has been integrated into model parameters, the system should support reliable rollback. Researchers should implement high-frequency checkpointing, which functions as version control for model weights, so that training can be replayed from a known-good state. If a source $S$ is later

determined to be unauthorized, the developer should avoid attempting to subtract it from the final model via unlearning. Instead, they should perform a surgical re-branching by reverting to the checkpoint $\theta_{t<\text{ingestion}(S)}$ from before $S$ was ingested and then retraining forward on a clean data path. This approach provides a concrete and auditable way to demonstrate that the model was produced without relying on the contested source. As this section does not address about curing past infringement but about what the industry should do going forward, given the diagnosis in Sections 4–7, this approach provides a practical solution that is feasible today since formally guaranteed unlearning at the scale of modern foundation models does not yet exist.

### 8.2. For Policymakers: Structuring the Compliance Market

Technical solutions are futile if the legal framework makes compliance prohibitively expensive. Policymakers must reform the market to ensure that obeying the law is cheaper and easier than stealing data.

- **Centralized Licensing via Data Trusts:** Navigating rights negotiations with millions of individual creators is transactionally impossible (the tragedy of the anti-commons). To solve this, regulators should establish Data Trusts or Collective Management Organizations.

  These entities would act as centralized clearinghouses, offering licenses on FRAND (Fair, Reasonable, and Non-Discriminatory) terms. Developers would pay a single standardized fee to the Trust, which then algorithmically distributes royalties to authors—potentially weighted by metrics like Data Shapley values. This eliminates the excuse that licensing is too hard while ensuring creators get paid.

- **Mandatory Rights Declaration (The Binary Beacon):** The current reliance on voluntary opt-outs (like `robots.txt`) is legally ambiguous and unfair to creators. We propose a statutory Binary Beacon regime: a mandatory, machine-readable signal that explicitly declares AI training status.

  Crucially, legislation must shift the default presumption from Opt-Out to Opt-In. Any content without a clear beacon should default to permission denied. This shifts the burden of verification away from the artist—who currently has to chase down scrapers—and places strict liability on the data acquirer to ensure they have explicit permission before ingestion.

## 9. Conclusion

**This paper has argued that post-hoc opt-out strategies do not, as a matter of law, eliminate liability arising from the training process.** Liability is not determined solely by a model's output; rather, infringement is established by the underlying acts of acquisition and reproduction. Once these unlawful acts are completed, technical adjustments to outputs cannot retroactively cure the illegality. This exposes developers to severe risks, including massive financial penalties and algorithmic disgorgement—the destruction of the model itself.

Our analysis confirms this through three key findings. First, unauthorized ingestion constitutes an incurable *completed act* ($t = 0$), and model weights function as *fixed copies* that ontologically preserve infringing value, rendering post-hoc opt-out legally irrelevant. Second, an *inescapable web of contract and tort liabilities* effectively overrides standard copyright defenses regardless of fair use claims. Third, the doctrine of *unjust enrichment* mandates the disgorgement of any algorithmic value derived from misappropriated foundations, denying the retention of illicitly gained benefits.

To avoid the risks, the industry must shift fundamentally toward *ex-ante process compliance*. Technologists must abandon opaque practices for architectures ensuring verifiable lineage, while policymakers must reconstruct markets to facilitate standardized licensing and clear consent. The focus must move from retroactively patching leaks to proactively securing the source.

Ultimately, the legitimacy of an AI system depends on the integrity of its inputs, not the safety of its outputs. True compliance is not about what a model *says*, but about how it came to *know*.

## Acknowledgements

This work is partially supported by JST CREST JP-MJCR21D1, JST BOOST JPMJBY24A6, and JSPS KAK-ENHI JP23K16940.

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

## A. Engineering Output Safety vs. Legal Reproduction Liability

### A.1. The Engineering Perspective: Output Equivalence and Functional Restoration

From an engineering standpoint, a model can be viewed as a function $f(x)$ that maps inputs to outputs. Compliance, on this view, is achieved by ensuring a safe output distribution. The dominant engineering logic is that even if a model $M$ has been trained on harmful data $D_{\text{harmful}}$, the concern is resolved if post-hoc techniques—such as unlearning, fine-tuning, or guardrails—can adjust the model's behavior such that it becomes statistically indistinguishable from a counterfactual model $M'$ trained without $D_{\text{harmful}}$. In this framework, the emphasis is placed on present functional safety rather than the historical lineage of how the model was trained.

### A.2. The Legal Perspective: Liability Becomes Fixed at the Moment of Reproduction

From a legal standpoint, by contrast, liability may arise not only from outputs but also from the act of reproduction itself during training. In many jurisdictions, once data is copied or stored—whether on disk or in memory—for training without authorization, the relevant infringement (or other violation) may already be considered a completed act.

Accordingly, post-hoc methods such as unlearning or output guardrails may reduce the risk of future harmful outputs, but they function only as prospective risk opt-out measures. They do not retroactively cure the unlawfulness of the earlier completed act of unauthorized copying or use. In other words, legal liability can become fixed at the point of reproduction, and post-hoc opt-out cannot, by itself, operate as a legal exculpation.

## B. Technical Taxonomy of Post-Hoc Opt-Out Methods

In this paper, we define the following two types of post-hoc opt-out techniques: inference-time control and parameter modification.

### B.1. Inference-Time Control (Prompt/Output Level)

This category includes approaches that constrain model behavior without changing the model's internal parameters. Because these measures are relatively inexpensive to deploy and can be updated quickly without retraining, they have become the dominant standard in commercial AI systems.

Safety filters operate as an external wrapper around a trained, frozen model. Systems such as Llama Guard (Inan et al., 2023) or OpenAI's moderation endpoints typically rely on separate, lightweight classifiers that monitor the interaction in real time. These components can detect and block prompts that request copyrighted material, and can also suppress, truncate, or rewrite outputs that appear to closely resemble protected works. Functionally, these techniques restrict what the user is allowed to see or receive, while leaving the underlying model weights—and any latent knowledge encoded in them—unchanged.

### B.2. Parameter Modification (Parameter Level)

This category aims to alter the model's internal state so that specific knowledge or capabilities are removed (or at least substantially weakened) within the weights themselves. Although academic interest in these methods is substantial, practical deployment remains difficult, in part because targeted edits can affect model stability and general performance.

**Machine Unlearning:** Machine unlearning methods seek to transform a model trained on dataset $D$ into one that approximates a counterfactual model trained on $D \setminus D_{\text{forget}}$ (i.e., the original dataset with a specified forget set removed).

- *Exact unlearning:* Methods such as SISA (Sharded, Isolated, Sliced, and Aggregated) (Bourtoule et al., 2021) can provide strong guarantees by retraining only the affected sub-models. However, at the scale of large language models, this approach is often computationally prohibitive.

- *Approximate unlearning:* As a result, most practical work focuses on approximations, such as targeted *gradient ascent* updates that attempt to undo the learning signal by reducing the likelihood of specific tokens or associations (Jang et al., 2023).

**Model Editing:** Model editing techniques attempt to locate where particular factual associations are represented in the

network and then modify those representations directly (Meng et al., 2022).

**Steering and Constitutional AI:** Approaches like *Constitutional AI* (Bai et al., 2022) train models to critique and revise their own outputs according to articulated safety principles. The result is an internalized refusal mechanism: the model may decline to generate infringing content in response to certain requests, even though the relevant information may still remain represented within the parameters.

Unlike inference-time guardrails, which restrict behavior at the inference, these methods aim to overwrite information within the weight matrices themselves, analogous to performing precise surgery to alter a specific memory.

## C. Glossary of Legal Terms

To bridge the epistemic gap between the machine learning and legal communities, this section provides concise, plain-language definitions of the core legal doctrines and terms invoked throughout this paper, as promised in the author response.

- **Ex-Ante:** Before the event. In this paper's framework, it refers to verifiable process compliance implemented at each pipeline stage before model production, ensuring that training data is legally clean and trackable from the start.

- **Post-Hoc:** After the fact. In machine learning engineering, this describes interventions or opt-out methods—such as inference-time guardrails or parameter modifications—applied after a model has already been trained.

- **Strict Liability:** A legal principle under which liability attaches automatically upon the completion of a prohibited act, regardless of the developer's subjective intent, knowledge, or negligence. In copyright law, once an unauthorized reproduction occurs, the statutory violation is finished, and subsequent technical patches cannot retroactively erase the accomplished fact.

- **Prima Facie:** A legal presumption that a statutory violation has been established based on the initial objective evidence, functioning similarly to a default "permission denied" status in software engineering. For example, downloading unauthorized datasets constitutes a prima facie violation of the reproduction right; it establishes a baseline of liability that stands as a historical fact unless the developer can successfully prove a valid affirmative defense, such as fair use.

- **Reproduction Right:** The exclusive statutory right granted to copyright holders to control the fixation of their protected works in copies. This right is implicated both during corpus building (downloading and storing data on servers) and training ingestion (loading datasets into volatile memory like RAM/VRAM).

- **Fixation:** The statutory requirement that a protected work must be embodied in a tangible, sufficiently stable medium to exist for more than a transitory duration. Model weights stored on physical media and computational copies residing in RAM during training both satisfy this legal requirement.

- **Browsewrap:** Website terms of service or agreements that users are legally deemed to accept simply by using or accessing the site, commonly utilized by online platforms to prohibit automated access, scraping, or data extraction.

- **Unjust Enrichment:** A legal doctrine requiring a party to surrender benefits obtained at another's expense without a lawful justification. In generative AI development, using protected content as computational scaffolding to avoid the immense costs of lawful data acquisition or licensing operates as an illicit shortcut that triggers this doctrine.

- **Disgorgement / Algorithmic Disgorgement:** A structural remedy designed to strip an infringer of wrongful gains by surrendering profits or benefits derived through wrongful conduct. In severe cases, this remedy requires reaching the model itself, mandating the complete deletion and destruction of the trained parameters to prevent the retention of distributed value.

- **Head Start Doctrine:** An equitable doctrine dictating that a defendant cannot retain a structural time-to-market acceleration or competitive advantage gained through improper shortcuts, even if they later deploy technical fixes or delete the original source materials.

- **Injunctive Relief / Injunction:** A judicial order commanding a party to do or refrain from doing a specific act—such as a court order to stop or mandate specific conduct—typically utilized to halt the deployment of a model or prevent ongoing unauthorized use when monetary exposure alone is insufficient.

## D. Brief Introduction to Copyright Basics for Technologists

To assist readers unfamiliar with copyright scholarship, this section provides an intuitive, example-driven introduction to the core concepts governing technical data usage under copyright law.

### D.1. The Lifecycle of Copyright: Fixation and the Reproduction Right

Copyright protection automatically attaches to original works of authorship the moment they are *fixed* in a tangible medium.

- **Example of Fixation:** Writing a poem on a piece of paper, saving a digital image to a hard drive, or storing text on a server all satisfy the fixation requirement. Conversely, an unrecorded live speech or a fleeting electrical signal does not.

- **The Reproduction Right:** This is the exclusive right of the creator to make copies of their fixed work. In common software engineering workflows—such as downloading an image corpus from the web onto a training cluster—a developer creates an unauthorized copy on their server, which legally constitutes a prima facie violation of this right.

### D.2. Affirmative Defenses: Fair Use and Transformative Use

When a developer reproduces a work without permission, they are legally liable unless they can establish a defense, most notably the doctrine of *fair use*. Fair use is an equitable balance test that excuses an otherwise infringing act based on four factors (purpose of use, nature of work, amount taken, and market effect).

- **Transformative Use:** Under the first factor, a use is considered transformative if it adds something new, with a further purpose or different character, altering the original work with new expression, meaning, or message.

- **The Search Engine Precedent (Complementary Use):** In *Authors Guild v. Google* (2015), creating a digital copy of millions of books to build a searchable snippet index was deemed a transformative fair use. This is because the search index served an entirely different purpose (finding books) and did not substitute for the original works; instead, it acted as a market complement.

- **The Generative AI Contrast (Substitutive Use):** As argued in Section 3 and 7, modern generative foundation models present a different paradigm. When a model ingests protected content to generate competing expressive text, code, or imagery, the ultimate use shares the same commercial purpose as the original data and acts as a high-tech market substitute, which we argue weighs against a fair use defense.

