# OpenReview forum: "Position: Infringement cannot be cured after training"
_ICML.cc/2026/Position_Paper_Track — ICML 2026 Position Paper Track spotlight_

### Official Review · Reviewer_kt3S · 2026-03-03

**Significance:** 4
**Argument Clarity:** 4
**Rating:** 6
**Confidence:** 5

**Questions:**

The picture you paint seems very clear... I understand that there is a ruling for Anthropic on Books3 but could you provide some commentary on why litigation isn't more widespread? (I understand that there are several court hearings in process but it seems like if infringement is this clear you would expect more litigation?)

414-421: If it was a formally gauranteed unlearning method then I don't see why you should prefer some reverse checkpointing method, in both cases (using your framework) the infringement is fait accompli. Both seem like clean and auditable (though I admit that since we don't have formally gauranteed unlearning then maybe your suggestion is of a practical flavour).

436-439 right column:  Perhaps there is some kind of specialized reading of "legitimacy" or "compliance" you have here but isn't this a counter example: if a generative model was trained on legally sourced material but is deployed for things that violate say the EU AI Act, wouldn't that consist of a non-legitimate model who is not compliant? Is there a subtle misreading in the text where the authors implicitly assume that a model trained on compliant data will itself be compliant? this is not true for generative models who can, for example, learn using in-context learning to do unsafe and illegal things regardless of how the model is trained.

**Alternative Views Section:**

Yes

**Compliance With Llm Reviewing Policy A Conservative:**

Affirmed.

**Discussion Potential:**

4

**Final Justification:**

The author has addressed my concerns.

**Paper Summary:**

This paper asserts that post-hoc interventions cannot remedy content infringement and other related legal injuries that occur before those interventions take place. The authors explain a variety of ML lifecycle phenomena from a legal perspective such as training as reproduction of content through corpus building or ingestion (fixation via RAM copies). They explain why many ML training and deployment practices might be explicit or implicitly breaching contract or be unfair. The present data as commons and transaction costs alternative views and a variety of engineering and policy solutions.

**Position:**

Yes

**Position In Title:**

Yes

**Related Work:**

3

**Strengths And Weaknesses:**

## Strengths

This paper is very well written and structured. The stated position is clear and clearly arguged for. The authors do an excellent and thorough job of tying current situations to  precedent (ProCD v Zeidenberg,  Mai v Peak, Microstar v FormGen). I believe this is a very important topic for the ICML community since much of current work is done using foundation models whose datasets may be in violation of the law. This paper may have a focusing effect because post-hoc solutions are another area of focus of the conference which may not have any legal relevance which calls their research motivations into question.

## Weaknesses

### Audience Awareness

My argument clarity score is slightly lower than I'd prefer to put only because, as a ML researcher, I had to look up many many terms (e.g., ex-ante). it might be nice to use the extra page if accepted to spend some more time considering your audience is unfamiliar with legal language. I think its important that you did use legal language since its important for the ICML community to understand legal thinking. Along these lines some knowledge of copyright is assumed, perhaps an appendix that gives some basic examples of fair and tansformative use, fixation, etc just a basic introduction to copyright so the text can be read easier by those without that background.

### Beyond Infringement

While I am fine with this paper primarily being about infringement, it seems natural to me that the paper could be enriched and extended quite easily to other types of unlawful acts that occur in training and deploying AI systems. The two I have in mind: illegal content like graphic violent and sexual content (e.g., the case of LAION deleting CSAM images) and privacy violations. It seems to me by adding these two additional categories you could make an even stronger case that post-hoc intervention are not suffeceint. This comment doesn't really effect my score because it might unduly impact the scope of your project.

## Comments (don't impact my score)

The running title isn't filled out
You should use \citet instead of e.g., Sag (Sag, 2018).
321-322 - I'm not sure that Ghorbani and Zou is a very good citation to use here. You might cite more more examples like the Phi series of models (Textbooks is all you need) or, for post-training, LIMA: Less Is More for Alignment. I am sure there are even more powerful examples that are for modern generative foundation models.

A very minor technical nitpick - 074-077 on the right column. Functional identity might be more powerful formalism because some AI systems are not always naturally formulated as conditional probabilities. For example, your arguments apply to an unconditional generative model (well i guess we can say x is empty) or a classification model (well I guess we can interpret logits probabilistically but technically that asserts a measure that we don't necessarily need).

**Support:**

4

---

> ### Author Rebuttal · Authors · 2026-03-30
>
> We sincerely thank the reviewer for the detailed comments and feedback.
>
> > ...I had to look up many many terms (e.g., ex-ante)...perhaps an appendix that gives some basic examples of fair and transformative use, fixation, etc...
>
> We agree. We will add an appendix with a brief introduction to copyright basics (fair use, transformative use, fixation, reproduction right, etc.) with concrete examples, and define legal terms such as "ex-ante" (= before the event) at first use.
>
> > ...the paper could be enriched and extended...to other types of unlawful acts...illegal content...and privacy violations...
>
> We deliberately scoped this paper to copyright because illegal content (e.g., CSAM) and privacy violations (e.g., under GDPR) each involve distinct legal frameworks that differ substantially from copyright law. That said, the intuition that a post-hoc fix may not undo a completed legal wrong could carry over; whether and how it does would require careful analysis grounded in each area's specific doctrines.
>
> > The running title isn't filled out. You should use \citet...You might cite...the Phi series...or...LIMA...
>
> We will fix the running title, use \citet consistently, and replace the Ghorbani and Zou citation at lines 321-322 with the Phi series (Gunasekar et al., 2023) and LIMA (Zhou et al., 2023), which more directly show the impact of high-quality data on foundation model performance.
>
> > ...Functional identity might be more powerful formalism because some AI systems are not always naturally formulated as conditional probabilities...
>
> Our argument does not depend on the model being a conditional probability distribution. We will revise the formulation to use functional identity, which covers unconditional generative models, classifiers, and other architectures without requiring a probabilistic interpretation.
>
> > ...could you provide some commentary on why litigation isn't more widespread?...
>
> Several factors explain this gap. Copyright lawsuits are costly, so only large, well-funded plaintiffs like the NYT and Getty Images have brought cases so far. The financial impact on individual creators is still emerging, making it hard to justify litigation costs. Key cases (Bartz v. Anthropic, NYT v. OpenAI, Getty v. Stability AI) remain ongoing, and the lack of precedent discourages new claims. We expect litigation to grow as harm becomes clearer, and will note this in the revision.
>
> > 414-421: ...I don't see why you should prefer some reverse checkpointing method...the infringement is fait accompli...maybe your suggestion is of a practical flavour.
>
> The reviewer is correct that the initial infringement is a fait accompli regardless of the remedy, and Section 8.1 does not contradict this.
>
> However, Section 8 is not merely disconnected from Sections 4-7. While reverse checkpointing cannot retroactively cure the completed act of copying (Section 4.1), it can concretely mitigate downstream liabilities that post-hoc methods cannot. Because this approach produces a new model whose parameters no longer embed the contested data's influence, it eliminates the head-start advantage discussed in Section 6.1, namely the persistent competitive acceleration that survives mere deletion of source files. By contrast, post-hoc mitigation leaves tainted parameters intact, meaning the developer retains the structural benefit of the original infringement. Reverse checkpointing is not a perfect defense against all liability layers in Sections 4-6, but it represents a materially stronger position than the post-hoc paradigm we critique.
>
> As the reviewer anticipates, our proposal is practical. Formally guaranteed unlearning at scale does not yet exist, so the industry needs a feasible solution today. Reverse checkpointing provides this: rolling back to the pre-ingestion state and retraining on clean data produces a lawful replacement without the prohibitive cost of retraining from scratch, which would otherwise incentivize keeping infringing models in place. Should guaranteed unlearning become available, comparing the two approaches would be valuable future work. We will foreground this rationale in the revision.
>
> > 436-439: ...if a generative model was trained on legally sourced material but is deployed for things that violate say the EU AI Act...the authors implicitly assume that a model trained on compliant data will itself be compliant?
>
> Our discussion in lines 436-439 is focused on training-stage compliance, i.e., whether the data was lawfully sourced. We do not suggest that lawful training data alone makes a model compliant in all respects. Deployment-stage risks such as in-context learning of unsafe behaviors and EU AI Act violations are distinct concerns outside our scope.

---

> > ### Author Rebuttal · Reviewer_kt3S · 2026-04-01
> >
> > Thank you for your reponse, I feel as though my questions / concerns are satisfactorily addressed - I have raised argument clarity scores. I appreciate the time taken both to write this paper and respond to my questions as I learned a lot about legal dimensions of foundation models and I am sure the community will to.

---

### Official Review · Reviewer_JjKf · 2026-03-11

**Significance:** 4
**Argument Clarity:** 4
**Rating:** 5
**Confidence:** 4

**Questions:**

Please see the weakness section.

**Alternative Views Section:**

Yes

**Compliance With Llm Reviewing Policy A Conservative:**

Affirmed.

**Discussion Potential:**

4

**Paper Summary:**

This paper studies the problem of copyright infringement and the corresponding post-hoc mitigation methods involved in generative AI models. The authors make a key argument that such mitigation methods cannot retroactively solve the infringement liability from unlawful data acquisition and training. The argument is threefold: (1) unauthorized acquisition itself can be a legally complete act, the model weights can operate as fixed copies; (2) licensed data should independently restrict access; (3) unjust enrichment and disgorgement. Finally, the authors raise a clear position on encouraging ex-ante process compliance.

**Position:**

Yes

**Position In Title:**

Yes

**Related Work:**

4

**Strengths And Weaknesses:**

**Strengths**

(1) This paper is very well written, and I thoroughly enjoyed reading it. The paper is nicely structured, and all claims and arguments are supported by evidence.

(2) This paper studies an important and timely problem of copyright infringement in generative AI, examining key mitigation methods and relevant legal discussions.

(3) This paper offers a strong and compelling position that will likely stimulate discussion in both the ML research community and the legal community.

(4) In my opinion, this paper addresses several long-standing open problems in this research field and exemplifies a strong position paper.

**Weaknesses**

(1) In Section 3, the authors mention that two vectors can be interpreted as reproduction, implying that acquisition and training themselves may serve as evidence of infringement. Does this suggest that post-training mitigation methods are rendered meaningless?

(2) In Section 4.2, the authors claim that weights are stable, non-transitory files. However, most machine unlearning algorithms (and model editing algorithms) do update model weights. Should this be considered an exception to fixation? Furthermore, if the data is not technically recoverable, does this imply that the mitigation is meaningful?

(3) It seems that exact unlearning and approximate unlearning could be treated quite differently in this context. For example, does SISA satisfy the architectural reversibility described in Section 8.1?

**Support:**

4

---

> ### Author Rebuttal · Authors · 2026-03-30
>
> We sincerely thank the reviewer for the detailed comments and feedback.
>
> >  (1) In Section 3, the authors mention that two vectors can be interpreted as reproduction, implying that acquisition and training themselves may serve as evidence of infringement. Does this suggest that post-training mitigation methods are rendered meaningless?
>
> For the purpose of curing the legal liability from training-stage infringement, yes, that is our argument. If data acquisition (Section 3.2) and training ingestion (Section 3.3) each constitute a completed act of infringement, the violation is already a legal fact before any post-training mitigation is applied. No subsequent action can undo a completed violation.
>
> That said, post-training mitigation is not meaningless in every sense. First, it remains valuable for addressing concerns beyond copyright infringement, for example, suppressing harmful or sexually explicit outputs, where the goal is not to cure a past legal wrong but to control the model's behavior going forward. Second, even within the copyright context, post-training mitigation may contribute to reducing the scope of damages a court might award, by limiting the extent to which infringing content is actually disseminated through outputs. Our claim is specifically that these methods cannot retroactively eliminate the liability that has already attached at the training stage. We will clarify this distinction in the revised manuscript.
>
> >  (2) In Section 4.2, the authors claim that weights are stable, non-transitory files. However, most machine unlearning algorithms (and model editing algorithms) do update model weights. Should this be considered an exception to fixation? Furthermore, if the data is not technically recoverable, does this imply that the mitigation is meaningful?
>
> Section 4 presents two distinct arguments, each targeting a different type of post-hoc mitigation (as categorized in Appendix B). Section 4.1 ("Completed-Act Doctrine") covers all post-hoc methods, regardless of type. Section 4.2 ("Ongoing Copy Status") focuses specifically on inference-time controls (Appendix B.1), which restrict model outputs without modifying the model itself.
>
> Because Section 4.2 only deals with inference-time controls, the model weights remain unchanged. Machine unlearning and model editing do modify weights, but they are simply not what Section 4.2 is about. So this is not an exception to the fixation argument; it is outside its scope.
>
> Regarding whether non-recoverability makes the mitigation legally meaningful: we argue it does not. This is where Section 4.1 applies. The core idea is that the infringement is complete the moment the unauthorized copying occurs. What happens to the copy afterward, whether it is preserved, modified, or effectively destroyed, does not change the fact that the infringement already took place. An analogy: shredding a pirated copy of a book does not undo the act of pirating it. In the same way, even if parameter modification renders the protected content no longer recoverable, the original act of unauthorized copying remains a completed wrong.
>
> We will revise Section 4 to make the respective scopes of these two arguments clearer, so that it is easier to see which type of mitigation each argument addresses.
>
> >  (3) It seems that exact unlearning and approximate unlearning could be treated quite differently in this context. For example, does SISA satisfy the architectural reversibility described in Section 8.1?
>
> We thank the reviewer for this insightful question. We think exact and approximate unlearning should indeed be treated quite differently in our framework, and SISA offers a helpful illustration of why.
>
> SISA shards data and trains independent sub-models, so when a source turns out to be unauthorized, only the affected shard needs to be discarded and retrained on clean data. When this architecture is adopted before training begins and the contaminated shard is genuinely discarded and rebuilt in a transparently auditable manner, SISA functions in much the same way as our "Git for Models" proposal in Section 8.1. Neither claims to cure the past infringement which remains a legal factm but both drastically lower the practical cost of producing a new, lawful replacement, which is exactly the problem Section 8.1 is trying to solve.
>
> Approximate unlearning, on the other hand, modifies the parameters of an already-trained model in place without discarding and rebuilding. We believe this falls within the kind of retroactive repair that Sections 4–7 argue cannot cure the completed act of infringement, and it therefore occupies a fundamentally different position in our framework.

---

> > ### Author Rebuttal · Reviewer_JjKf · 2026-04-01
> >
> > Thank you for the detailed rebuttal! The discussions were very interesting, and the authors' responses align well with my intuitions. I encourage them to incorporate these clarifications into the final draft to further strengthen the paper.
> >
> > I remain very positive about this work. If published, I believe it will stimulate meaningful and important discussions in the field.

---

### Official Review · Reviewer_jQAE · 2026-03-12

**Significance:** 3
**Argument Clarity:** 1
**Rating:** 3
**Confidence:** 4

**Questions:**

Does "U.S.C." mean U.S. Court? Define it.

There are many problematic cites, e.g., (nyt, 2023) (get, 2023) (tex, 1994)

**Alternative Views Section:**

Yes

**Compliance With Llm Reviewing Policy A Conservative:**

Affirmed.

**Discussion Potential:**

2

**Final Justification:**

I've adjusted the score to 3, though I still have concerns about how to view and benefit from the position paper, which is vague yet could be arguably subjective.

**Paper Summary:**

This paper argues that such post-hoc mitigation strategies do not, as a matter of law, eliminate legal liability arising from the training process.

**Position:**

Yes

**Position In Title:**

Yes

**Related Work:**

1

**Strengths And Weaknesses:**

Strengths:

+ There is a gap between legal and technical practice in copyright protection.

+ The paper tries to decompose the processing steps using data and reason the causal for each step.

Weaknesses:


- It is hard to follow the paper. For examples, just in abstract:
  - What are "post-hoc mitigation methods". The concept is very broad, especially when not enough context are provided on the issues to be mitigated.
  - "Post-Hoc Sanitization" is not introduced.
  - "Ex-Ante Pro- cess Compliance" is not defined.
I'd suggest reconstruct the paper to create a clearer flow of the arguments, solutions and debate with the alternative views. Many points are mixed with others.



- The suggested actions are not systematic:
  - What is the connection between those to engineers and policy makers? How can they coordinate?)
  - What is the connection between "Glass Pipeline" and "Architectural Reversibility"?

- The paper proposed alternative views and responses to them, though some responses can be extended and included to this work as contributions.

**Support:**

1

---

> ### Author Rebuttal · Authors · 2026-03-30
>
> We sincerely thank the reviewer for the detailed comments and feedback.
>
> > What are "post-hoc mitigation methods". The concept is very broad, especially when not enough context are provided on the issues to be mitigated.
>
> In lines 31-38, post-hoc mitigation methods are defined as “interventions applied after a model has already been trained,” such as opt-out mechanisms and machine unlearning. In this paper, we focus on mitigating issues related to copyright infringement. The taxonomy of post-hoc mitigation methods is summarized in Appendix B, which will be moved into the main body in the camera-ready version.[1](https://docs.google.com/document/d/1l7VO6ZdFDnje4Am5AU3QolDdVpHnPQssD4ZFukmBmHk/edit)
>
> > "Post-Hoc Sanitization" is not introduced.
>
> > "Ex-Ante Pro-cess Compliance" is not defined.
>
> > Does "U.S.C." mean U.S. Court? Define it.
>
> We apologize that these terms are not clearly defined. "Post-Hoc Sanitization" is a paraphrase of “post-hoc mitigation”. "Ex-Ante Process Compliance" is legal compliance at each pipeline stage before model production, such as removing all harmful content from the training dataset before training. "U.S.C." stands for the United States Code. At the camera-ready stage, we will check that every term is defined at its first occurrence and used consistently throughout the paper
>
> > I'd suggest reconstruct the paper to create a clearer flow of the arguments, solutions and debate with the alternative views. Many points are mixed with others.
>
> Each section plays its own role. In Section 3, as a premise of the discussion, we argue that unauthorized training constitutes copyright infringement. In Sections 4–6, we discuss why post-hoc mitigation cannot cure copyright infringement. We debate the alternative views in Section 7 and provide solutions in Section 8.[1](https://docs.google.com/document/d/1l7VO6ZdFDnje4Am5AU3QolDdVpHnPQssD4ZFukmBmHk/edit)
>
> While this roadmap appears in the introduction (lines 66–86), we will clarify it to make the logic visible at a glance.
>
> > The suggested actions are not systematic:
>
> > What is the connection between those to engineers and policy makers? How can they coordinate?)
>
> > What is the connection between "Glass Pipeline" and "Architectural Reversibility"?
>
> Thank you for pointing out the need to clarify this connection. These concepts operate together as a unified system.
>
> Engineers and policy makers form a mutually dependent compliance system. Policymakers must establish legal mandates to make compliance binding, while engineers must build the technical tools to make those mandates executable.
>
> The Glass Pipeline acts as a diagnostic tool to precisely detect when unauthorized examples are ingested, while Architectural Reversibility provides the mechanism to surgically undo that specific training step on those examples.
>
> We will add a brief paragraph in Section 8.1 to explicitly connect these complementary concepts.
>
> > The paper proposed alternative views and responses to them, though some responses can be extended and included to this work as contributions.
>
> Thank you for the suggestion. We agree that the current framing of Section 7 is overly defensive. Since each subsection introduces a novel perspective, we will expand on these points and reposition them as contributions in the revised manuscript.
>
> > There are many problematic cites, e.g., (nyt, 2023) (get, 2023) (tex, 1994)
>
> Thank you for pointing this out. We will correct the BibTeX entries in the revised manuscript so that legal cases are properly cited.

---

> > ### Author Rebuttal · Reviewer_jQAE · 2026-04-03
> >
> > I've raised my score, though I still have several concerns and pessimistic to the submission, mostly because (1) there is a gap in writing and organization to clearly understand the position, suggestion and debate logically; and (2) my research is related to this area, though I cannot easily get hints by reading this position paper, though I agree this feeling is arguable and subjective.

---

### Official Review · Reviewer_QL5y · 2026-03-14

**Significance:** 4
**Argument Clarity:** 3
**Rating:** 5
**Confidence:** 2

**Questions:**

Could the authors further clarify the distinction between the contract-law violation precedents discussed in Sections 5.1–5.2 and the issues of source legality and opt-out mechanisms described in Section 3? Specifically, does the difference hinge on whether the data itself is subject to copyright protection?

**Alternative Views Section:**

Yes

**Compliance With Llm Reviewing Policy A Conservative:**

Affirmed.

**Discussion Potential:**

4

**Final Justification:**

This manuscript explicitly discusses the tension between data and foundation model training from a legal perspective. During the rebuttal, the authors addressed almost all of my concerns. I hope the authors will revise and enrich the manuscript accordingly, so that the work can better benefit the machine learning community. I maintain my positive attitude toward this submission.

**Paper Summary:**

This position paper examines the legal tension between data acquisition/ingestion practices in current AI model training pipelines and their potential to constitute copyright infringement. The authors argue that developers should shift focus from ensuring harmless outputs to scrutinizing the legality of data lineage itself.

Through the lens of copyright law, contractual doctrines, and tort/unfair competition theories, the authors contend that current data utilization paradigms in AI training give rise to non-retroactively-curable infringement.
They argue the liability attaches at the moment of copying and cannot be undone by post-hoc mitigation techniques such as machine unlearning or inference-time guardrails. The authors further address counterarguments from the ML community and conclude with concrete recommendations for both AI developers and policymakers toward verifiable Ex-ante Process Compliance.

**Position:**

Yes

**Position In Title:**

Yes

**Related Work:**

3

**Strengths And Weaknesses:**

- Strengths
    - The tension between training data practices and legal accountability is highly relevant to the ML community. It deserves serious attention, particularly given the rapid development of diverse foundation-model-based AI systems.
    - The paper grounds its argument in a rigorous cross-jurisdictional legal framework, drawing on case law from the US, EU, and Japan to support its claims. The use of concrete judicial precedents strengthens the credibility of the doctrinal analysis.
    - The authors try to provide a practical roadmap toward Ex-ante Process Compliance.

- Weaknesses
    - As ICML is a machine learning venue, readers may find the density of legal terminology a barrier to engagement. Some key doctrinal concepts, such as "injunctive relief", "disgorgement", and "browsewrap", would benefit from brief, accessible definitions, either in the related work section or a dedicated section in the appendix.
    - Some legal precedents cited in the paper, such as MAI Systems (1993) and ProCD (1996), were decided in different technological contexts. More recent rulings from the rapidly evolving generative AI era, particularly those after 2023, would better substantiate the proposed claim.

**Support:**

3

---

> ### Author Rebuttal · Authors · 2026-03-30
>
> We sincerely thank the reviewer for the detailed comments and feedback.
>
> >  As ICML is a machine learning venue, readers may find the density of legal terminology a barrier to engagement. Some key doctrinal concepts, such as "injunctive relief", "disgorgement", and "browsewrap", would benefit from brief, accessible definitions, either in the related work section or a dedicated section in the appendix.
>
> We agree with this point. In the revised manuscript, we will add a glossary appendix with plain-language definitions of key legal terms (e.g., injunctive relief = a court order to stop or mandate specific conduct; disgorgement = surrendering profits gained through wrongful conduct; browsewrap = website terms that users are deemed to accept by using the site). We will also define each term briefly at its first use in the main text.
>
> >  Some legal precedents cited in the paper, such as MAI Systems (1993) and ProCD (1996), were decided in different technological contexts. More recent rulings from the rapidly evolving generative AI era, particularly those after 2023, would better substantiate the proposed claim.
>
> We acknowledge that relying on older precedents without sufficient context may create a disconnect for readers. The current manuscript does cite several post 2023 rulings — Bartz v. Anthropic (2025), Kneschke v. LAION (2024), Thomson Reuters v. Ross (2023), and Andy Warhol Foundation v. Goldsmith (2023) — but we should have done a better job of foregrounding them. Additionally, as the reviewer may be aware, many of the major AI copyright cases, such as NYT v. OpenAI and Getty v. Stability AI  are still in progress, and definitive rulings on several key issues have not yet emerged. Given this still developing landscape, we also relied on older precedents like MAI Systems and ProCD for the legal principles they establish. However, we recognize that we did not make this rationale sufficiently clear. In the revised manuscript, we will better explain the connection between these older principles and current AI litigation, and explicitly note the evolving state of AI-specific case law.
>
> >  Could the authors further clarify the distinction between the contract-law violation precedents discussed in Sections 5.1–5.2 and the issues of source legality and opt-out mechanisms described in Section 3? Specifically, does the difference hinge on whether the data itself is subject to copyright protection?
>
> The distinction is about the legal basis, not the data type.
>
> Section 3 addresses copyright infringement: the unauthorized reproduction of copyrighted works during dataset creation and training. The relevant question is whether the data is protected by copyright and whether an exception (e.g., fair use) applies.
>
> Sections 5.1–5.2 address contract-based claims: violations of license agreements or website terms of service. These are independent grounds of liability that arise from the breach of contractual terms governing access and use, regardless of the data's copyright status. Crucially, copyright and contract claims can coexist for the same data: a developer who accesses copyrighted content through a website that prohibits scraping may face both copyright infringement liability (Section 3) and contractual liability (Section 5) simultaneously.
>
> The practical consequence is that even if a developer could establish a copyright defense (e.g., fair use), they may still face liability for breaching the contractual terms under which they accessed the data.

---

> > ### Author Rebuttal · Reviewer_QL5y · 2026-04-02
> >
> > I appreciate the authors' response and their explicit discussion of the tension between data and foundation model training from a legal perspective. I hope the authors will revise and enrich the manuscript accordingly, so that the work can better benefit the machine learning community. I remain positive about this submission.

---

### Decision · Program_Chairs · 2026-04-30

**Decision:**

Accept (spotlight)

**Comment:**

The paper addresses a critical and timely issue at the intersection of law and machine learning that has potential to influence further technical work in the community and is likely to stimulate important discussions. Reviewers found that claims and arguments made in the paper are well-supported by evidence, and the paper is generally very well-written.

The primary concern raised by reviewers, especially Reviewer jQAE, was that the density of legal terminology makes it difficult for the technical audience of ICML to engage with the paper. The authors promised to address this issue in the rebuttal by committing to include a glossary of terms in the appendix, with clear definitions.

During the rebuttal, the authors also discussed other issues, including ongoing legal cases, and the difference in technological contexts in which previous cases were decided, nuances between exact and approximate unlearning and their implications, and the extension of claims beyond just infringement to include other types of unlawful acts relating to the training and deployment of ML models. Reviewers signaled that the rebuttal comprehensively addressed their concerns with the exception of clarity concerns of Reviewer jQAE. However, I feel that these can be mitigated to a large extent in the camera ready via the adjustments that the authors have committed to making.

Overall, the paper clearly states and sufficiently supports the critical and timely position that post-hoc mitigation strategies (such as machine unlearning and inference-time guardrails) are legally insufficient to "cure" liability arising from unauthorized data acquisition and training. This is an important discussion topic as it may influence technical research directions, so I agree with reviewers that this work deserves attention from the community and I recommend acceptance.